# The Secret Lives of Bouki: Louisiana's Creolized Folkloresque

**Rich Paul Cooper**

English Department, College of Liberal Arts and Sciences, Texas A&M University, College Station, TX 77843, USA; rcooper1@tamu.edu

**Abstract:** This article historicizes the character of Bouki in the context of Creole Louisiana, showing how the story of Bouki has evolved to become the story of Kouri-Vini, Louisiana's native and endangered Creole language. This historicization takes place in three distinct periods; those periods are defined by their relation to Kouri-Vini. The first period aligns with the Antebellum period; the second aligns with the early 20th century; and the final coincides with the present day. Moving across these periods, Bouki finds himself demoted, at which point he enters the 'creolized folkloresque.' The folkloresque is a larger mosaic of folkloric forms detached from the material conditions of their production and available to popular culture; for the folkloresque to be creolized designates the same process but under vastly unequal social and material conditions. In short, Bouki enters the creolized folkloresque, becoming a folkloresque figure available to all who find themselves subject to creolized conditions. In the pre-American part of Louisiana's history, creolized conditions included slavery and colonization; post-Americanization, linguistic discrimination plays an outsized role. Where such conditions persist in Louisiana, there Bouki can be found.

**Keywords:** Bouki; Louisiana; Kouri-Vini; Creole; French folklore; creolization; folkloresque

---

## 1. Bouki in Creole Louisiana





*Tou bouki-yé pa kanay*
*sé gin pèr vyé lapin-yé.*

*All unsuspecting critters*
*should fear wily rabbits.*[1]

Louisiana folklore offers a rich array of figures and stories drawn from many different cultures. While most of these stories are available to the anglophone reader, the true variety and nuance of those stories are only available to the francophone or creolophone reader. This trilingual readership points to the complicated colonial legacy of Louisiana, which takes place in two phases. In the first phase, as part of the French and then the Spanish empires, Louisiana was a settler-slave state composed of populations from many different cultural backgrounds. Unlike slavery in the British colonies, early Louisiana settlers sometimes married across racial and cultural lines. Even if such intermarriages were not common, Louisiana's cultural heritage exhibits African, European, and Native American roots. Perhaps nowhere is this mixture more evident than in Louisiana Creole or Kouri-Vini (LC or KV hereafter)[2]. In the second phase of Louisiana colonialism, Louisiana was colonized by American, English-speaking forces. Though it is difficult to put a date on the exact end of this period (certainly post-WW2), this period begins quite clearly with the Sale of Louisiana in 1803[3]. During this period, to be francophone or creolophone was to exhibit an identity and community directly at odds with *les 'Méricains*, or the Americans.

Even today, linguistic identity remains a powerful point of contention in Louisiana. After WW2, more and more French and KV speakers became 'American,' dropping their heritage languages because of the social stigma and shame attached to them. Instead of being passed on to the next generation, the languages became the secret tongues of the older

generations[4]. Consequently, though in the 1970s there were more than 1 million French speakers in Louisiana, by the turn of the century, those numbers had dwindled to less than 200,000 (Valdman and Picone 2005). Yet, aided by immersion schools, grassroot media, and educational campaigns, French is experiencing a modest renaissance in Louisiana. By virtue of its contact with the international Francophonie, Louisiana French is able to reach a wider audience than KV. This dialect is sometimes called Cajun French, but that is a bit of a misnomer. In reality, there were four French-speaking groups in Louisiana: white Creoles, Creoles of color, Francophone Amerindians, and Cajuns (Trépanier 1991). If the word Cajun is meant to designate the descendants of the Acadians who fled Canada during *Le Grand dérangement*, then Cajun French could only be said to designate a portion of the varieties of French found in Louisiana. Native American communities spoke and still speak French, as do Louisianans with African, Spanish, and even German heritage. In places such as New Orleans and among gentrified Cajuns, 'Plantation Society French' would have been as much in vogue as the types of French heard up and down the bayous of south Louisiana (Picone 2015, pp. 267–87). To make matters more complicated, many self-identified Cajun families actually speak KV (Dajko 2012). While many white Louisianans such as Alcée Fortier and Alfred Mercier learned KV from their enslaved nurse-maids, other white Creoles would come to speak this uniquely New World language as their mother tongue without ever having owned any slaves. As these different groups spoke this language, this New World, Afro-Indigenous language, they (re)transmitted old ideas and stories in a radically changed colonial context.

KV has been gaining momentum, too, even if it occupies a more precarious position than Louisiana French. In "The Louisiana Creole Language Today" (Klingler 2019), Thomas Klingler offers an excellent summary of the research and activist work being performed in KV up until the date of publication. Since the publication of his article, language activist Taalib Pierre-August has self-published the first novella in KV and a collection of essays (Pierre-August 2023a, 2023b), the first collection of poetry in KV (Mayers 2022), and the first film (Harris 2023). Some organizations are at work in the arts, such as Opera Créole (Opera Créole 2024), while others are at work in language activism, such as Chinbo (Chinbo 2024). Language classes in KV are offered by Chinbo, L'Alliance Française of New Orleans and the Nous Foundation. There are even enough speakers and writers of KV to warrant Éditions Tintamarre at Centenary College hosting a category for KV stories during a 2023 competition for the best fairy tales (*contes merveilleux*), and for artist–activist Jonathan Mayers to host a multi-artist, multi-writer, multi-lingual 2023–2024 exhibition at the Capitol Park Museum in Baton Rouge, Louisiana (Mayers 2023–2024). Klingler concludes his summary by stating that KV "is today both one of the most seriously endangered creole languages and among the most thoroughly described" (Klingler 2019, p. 94). While the prognosis may yet be grim for KV, it seems that the current speakers are intent on leaving a rich body of art and literature for future scholars to describe and translate.[5] Louisiana French certainly has the tactical advantage over KV in this arena. There is not, as such, an international Creolophonie. Within the Creole-speaking world, there exists, after all, a good deal of difference between the Creole languages from different places such as Haiti, Guadalupe, or Martinique.

Such heterogeneity defines Creole-ness, such that creolization is often seen as a process of 'mixing.' Following this logic, creolization is sometimes applied to phenomena outside the sphere of any historically situated '*créolité*.' This article will use the concept of creolization in its historical sense because KV is a particular, historically located phenomenon emerging from the complex history of Louisiana. Emphasis must be placed on the historical embeddedness of creolization to distinguish it from Édouard Glissant's conception of creolization. According to Richard Price, in his article, "*Créolisation*, Creolization, and *Créolité*", *créolisation* in the Glissantian sense is "a state of being, an essence—not, like historical creolization, a process" (Price 2017). *Créolisation* following Glissant is a larger concept, one capable of describing the situation of modern-day Lagos or Singapore, any place where different identities, languages, and cultures are in constant contact with each other, naturally

mixing as a result. *Creolization*, conversely, emphasizes the historical material conditions of slavery under which this process happened, emphasizing the ways that "diverse societies and cultures are suddenly thrust together and create new social and cultural institutions under conditions of vastly unequal power" (Price 2017). Creolization in this article is used, then, always in a sense cognizant not only of the unequal power distributions under slavery and colonialism in general, but also of the unequal power distributions of slavery and colonization specifically as they have manifested in Louisiana.

Because enslavement and colonization conspired to limit KV to oral expressions, the shift from oral folklore to written literary fairy tales should be met with no small amount of celebration. This shift provides a unique opportunity for the student of folklore and fairy tales to bear witness to a major shift in form, a remarkable shift that has gone relatively unnoticed in most major accounts of Louisiana folklore. This article follows closely in the footsteps of Jack Zipes, whose life's work in folklore and fairy tales has been to elucidate the changing material conditions that brought about the new literary form we call the fairy tale. Zipes writes, "the tales are reflections of the social order in a given historical epoch, and, as such, they symbolize the aspirations, needs, dreams and wishes of common people in a tribe, community, or society, either affirming the dominant social values and norms or revealing the necessity to change them" (Zipes 2002, p. 7). As the oral folktale becomes the literary fairy tale, the stuff of the original folklore becomes available to a wider audience, joining what Michael Dylan Foster calls 'the folkloresque' (Foster and Tolbert 2016). This concept usefully describes how specific folklore traditions or figures detach from their origins and become part of the mosaic of popular cultures. An excellent example would be literary fairy tales, which were first written by educated and refined folk such as Madame D'Aulnoy in the 17th century. To illustrate how the development of the literary fairy tale is central to the development of the folkloresque, consider the following quote by Foster. Regarding the folkloresque, he informs us that the folkloresque is "consciously cobbled together from a range of folkloric elements, often mixed with newly created elements, to appear as if it emerged organically from a specific source" (Foster and Tolbert 2016, p. 5). While Foster here describes 'the folkloresque,' this quote could also be used to describe the work of early literary fairy tale writers such as D'Aulnoy, who organically cobbled together a range of folkloric elements with new literary elements. This confluence indicates that the development of the literary fairy tale is an important step in the formation of modern folkloresque.

This article will historicize the shift from oral folktales to literary fairy tales in the context of Louisiana through the character of Bouki, whose unique origins in Africa and subsequent introduction and perpetuation into present-day Louisiana depict shifting ideologies shaped by slavery, colonization, assimilation, and language loss. In modern Louisiana, the word bouki is most commonly translated as hyena. For a more complete exploration of the significance and portrayal of the character of Bouki across Africa, see Marcia Gaudet's article "Bouki, the Hyena, in Louisiana and African Tales" (Gaudet 1992). There, Gaudet highlights the sexually ambivalent nature of the hyena, but she also makes clear Bouki's status as a 'duped trickster.' According to Gaudet, "Although Lion and Elephant are favorite dupes, Bouki/Hyena is often taken in by (Hare) Rabbit/Lapin to pursue trickster behavior, but he is almost never successful" (p. 70). In Louisiana stories, Bouki's status is made more liminal by virtue of the fact that most representations of Bouki are mediated, interpreted, translated, and re-made by white Creoles. In the first recordings of the Louisiana tales by Alcée Fortier, Fortier translates Bouki as goat, a translation no doubt signaling how far Bouki had traveled from his origins. In Louisiana, there are no hyenas after all. As Bouki entered more and more into the 'folkloresque' mosaic of Creole Louisiana, he often lost his animal identity altogether, instead functioning as a generic dupe or fool character, widely referred to by the sobriquet 'Ol' Tonton.' However, with the advent of the modern literary fairy tale in Louisiana, which has only existed in French and KV since 2023, a radical change occurs. White Creoles and Creoles of color begin telling stories in conjunction with each other and with people from around the world, but especially Africa. Bouki is no

longer a dupe, but a wise friend; even wily Lapin changes his ways. Simply put, early retellings of the Bouki stories focused on unequal power structures such as enslavement and colonization, while modern retellings focus on linguistic struggles such as the work of keeping KV alive.

Following Zipes' logic, to historicize the character Bouki in Louisiana folklore is to reveal not only shifting political categories and identities but also changing assumptions about the nature of reality, community, language, and identity. Bouki, whose name means hyena in the Wolof language of Senegal, travels to the New World in the mouths of the enslaved, persists in Louisiana among white Creole communities, and nearly disappears altogether before making a cameo in one of the first *contes merveilleux*—literary fairy tales— written in KV (Mayers and Johnson 2023). Bouki provides, then, a unique perspective on the shifting and hybrid contexts of Creole Louisiana. To help conceptualize these shifting and hybrid contexts, this article relies upon the chronotopic schema laid out by N.A. Wendte in his 2022 essay "The Chronotopic Organization of Louisiana Creole Ethnolinguistic Identity". For the purposes of this argument, chronotopes can be seen as sociohistorical snapshots in time; Wendte defines three in Louisiana vis-à-vis KV: the Purity chronotope, the Authenticity chronotope, and the Revitalization chronotope. All three periods happen after the Civil War, only engaging with colonial Louisiana as a mythic past. The final period is the current day, after the campaign of Americanization has successfully assimilated most Louisiana Creoles. All three periods will be defined further in the sections below, with the periods serving as scaffolding and guides. This choice to focus on the linguistic chronotope is not meant to detract from or ignore other sociohistorical elements at play. Rather, since language has played such a large part in Louisiana Creole identity, to trace Bouki across these periods is to link Bouki to a larger narrative of economic, cultural, and linguistic erasure. In short, the story of Bouki in Louisiana has become the story of the Creole people and their language, Kouri-Vini.

## 2. Bouki in the 'Purity' Chronotope

During the Purity period, the recorded Bouki tales were told by those formerly enslaved to white Creole folklorists who worked to transcribe and translate those stories for an Anglophone audience. Wendte dates the 'Purity' period from the Civil War and through Reconstruction, emphasizing purity because white Creoles of the Antebellum period were obsessed with the idea of Creole purity (p. 100).[6] In the analysis of Alcée Fortier's collection of stories that follows, it must be emphasized that while these stories are sold as authentic transcriptions of slave recordings, they are not. Accordingly, they are treated below as white performances of blackness, following Jennifer Gipson in her essay "'A Strange Ventriloquous Voice': Louisiana Creole, Whiteness, and the Racial Politics of Writing Orality" (Gipson 2016). Gipson subverts the common notion that Fortier's voice should be "equated with the black Creole-speaker's voice" (p. 461). In fact, it is not until the Revitalization period that we hear a black Creole speaker's voice unmediated by white interlocuters.

The original KV language version of the briar patch comes from Alcée Fortier's 1895 collection *Louisiana Folk-tales*. A white Creole raised by caretakers who spoke Kouri-Vini, Fortier, had direct contact with formerly enslaved Creoles of color. However, it is not without some disdain that Fortier describes those whose stories he has taken the responsibility to share; he writes, "one must bear in mind that most of [these tales] were related to children by child-like people" (Fortier 2011, p. xvii). The children in question are, of course, white children, and the child-like people are Creoles of color, or in Fortier's own words, "a race rude and ignorant" (Fortier 2011, p. xvii). Without speculating too wildly, it seems apparent that the stories the enslaved shared with their young masters were perhaps sanitized for consumption by their young charges, stories suitable enough to pass the hasty check of some overbearing mother or father, innocuous enough to be re-told by naïve children. More tellingly, Fortier admits that Creoles of color were not eager to share stories with him: "It is a strange fact that the old negroes do not like to relate those tales with which they enchanted the little masters before the war" (Fortier 2011, p. xvii). If

the trickster stories, such as the tar-baby, are folkloric retellings shaped by the pressures of enslavement, then it should not be surprising to suggest the tellers related to the tales differently once slavery ended. Fortier was also a white supremacist. I do not intend to cast Fortier as an entirely unreliable ethnographer. After all, his story "Fillèle Compair Lapin" (Fortier 2011, p. 32), which involves Lapin escaping into the briar patch, is a near-exact re-telling of "Brer Rabbit and the Briar Patch". (ATU 175, 'The Tar-Baby and the Rabbit'). Despite such seeming fidelity, one must still wonder what tales the formerly enslaved decided to keep secret.

Rather than framing the problem as some individual problem on the part of Fortier, it is better to frame it as a problem of opposing cultural worldviews. Rabalais hits upon these opposing worldviews when analyzing the difference between African folklore and Louisiana folklore. In Louisiana, "the implication of a greater society (villages, families, tribal origins)" and "the relation between the individual and the community" are both lost, in a manner indicating the traumatic breaking of the social order suffered by the kidnapped and enslaved (Rabalais 2021, p. 33). Rabalais cites Koffi Konan, a scholar of African folklore, to emphasize the integration of folklore into the society, with folklore serving as an important tool for learning how to behave in society. Rabalais goes on to recognize the fundamentally different worldview found in relation to Louisiana stories, such that in Louisiana, "animal tales seem to exist in a spatial and temporal frame removed and independent from the real world", unlike the African tales, where the figures are integrated into reality itself (Rabalais 2021, p. 33).

The distinction drawn here by Rabalais deserves more attention. In Africa, the folklore figures are integrated into reality itself. This implies a level of belief missing from the Louisiana versions of the same tales. Indeed, this level of belief would warrant calling these tales legends or myths, following the classifications made by W. Bascom in "The Forms of Folklore: Prose Narratives" (Bascom 1965). According to Bascom, "myths are prose narratives which, in the society in which they are told, are considered to be truthful accounts of what happened in the remote past" (p. 4). Legends are "prose narratives which, like myths, are regarded as true by the narrator and his audience, but they are set in a period considered less remote, when the world was much as it is today" (p. 4). Conversely, folktales are "prose narratives which are regarded as fiction" (p. 4). However, it is not that Rabalais has made a categorical error. Rather, Bascom concedes that these distinctions are irrelevant to folklorists working in tale identification or historical-geographical methods (Bascom 1965, p. 7). However, for the purposes of this historicization these distinctions are paramount.

When, then, did Bouki cease to be integrated into reality itself? Slavery seems the obvious and easy answer. *Christianity* was certainly partly to blame, everywhere demoting the 'idolatrous' legends and myths of the people to the status of fiction only fit for children. In this religious sense, the peasant of the English countryside was under similar pressures as the enslaved Creole. About the English in the 18th century, Joseph Addison remarks, "There was not a village in England that had not a ghost in it, the churchyards were all haunted, every large common had a circle of fairies belonging to it, and there was scarce a shepherd to be met with who had not seen a spirit" (Addison 1712). Such superstition would be a direct threat to religion, whether it came from the English peasant or the enslaved African. Not only did Christianity look askew at such beliefs, so too did the general Enlightenment attitude of the West, an attitude that actively sought to eradicate legend and myth under the guise that such material was harmful to the developing minds of children. As early as 1692, John Locke expressed this opinion in "Some Thoughts Concerning Education". Regarding an imagined pupil, Locke writes the following:

> Whilst he is young, be sure to preserve his tender mind from all impressions
> and notions of spirits and goblins, or any fearful apprehensions in the dark.
> This he will be in danger of from the indiscretion of servants. . .. Such bug-bear
> thoughts once got into the tender minds of children, and being set on with a strong

impression from the dread that accompanies such apprehensions, sink deep, and fasten themselves so as not easily, if ever, to be got out again. (Locke 1692)

Once the enslaved Africans had assimilated into Christianity, their original folktales would have been immediately quarantined as "superstitions" that could be dangerous to children. Under the dual pressures of the Church and the Enlightenment, a similar movement away from legends and myths swept the Western world. The difference between the situation of the enslaved and the situation of the peasant here might best be encapsulated by the fact that the former was considered a child while the latter was not.

Racial disdain is quite evident across Fortier's 1895 collection, only furthering speculation that the enslaved people might have censored the kinds of stories they told for their young white masters. In most stories featuring Bouki, Bouki plays the dupe to Lapin's trickster. This is the case in stories "Compair Bouki et Compair Lapin, Nos. 1–6" from part three (Fortier 2011, pp. 138–45). In these stories, Bouki and Lapin are portrayed as sort of "frenemies", friends who share similar goals yet will not hesitate to dupe the other in order to come out ahead. In these tales, Bouki is beaten, burned, stuffed with sand, ridden like a horse, robbed, and whipped. Only in one case does Bouki die, when the cork holding in the sand with which he has been stuffed is removed by his children, and even then, in comic fashion reminiscent of Wile E. Coyote and the Roadrunner. This series of tales ends with the admonition, "All the goats (*bouki*) which are not rascals/ought to fear the old rabbits" (Fortier 2011, p. 145). Rabalais notes that in Louisiana, these folktales are not part of the social fabric in the same way as they were in Africa; while that is the case, it may be more accurate to say that the tales have shifted to become part of a new social fabric, a European social fabric where such tales were considered, at best, charming fictions. If we take these six tales to be didactic tales for *white* children told by caretakers of color, then Fortier's couplet indicates that Bouki is the character with whom the children are asked to identify. Any little hyena can be good or bad, it would seem, but old rabbits are to be feared. Whereas in Africa, the trickster figure such as Lapin would be punished and the social order restored, Lapin escapes all such punishment in Fortier's tale. If read in this way, "old rabbits" could be said to represent the white parents of the little masters to whom these stories would have been told. It would not be considered, after all, a stretch to say that their parents used trickery in service of greed to overturn the social order and escape punishment. Little unsuspecting critters must keep an eye on wily rabbits like those.

Bouki appears in three other stories from Fortier's collection, acting as either the dupe, the trickster, or both at once. In the tale "Compair Bouki and the Monkey", Bouki tricks a group of monkeys into his boiling pot before consuming them (Fortier 2011, pp. 24–26). One of the monkeys escapes, warning his brethren about Bouki's tricks. When Bouki attempts to trick the monkeys, he is captured in his own trap and boiled alive. In the tale "Compair Bouki, Compair Lapin, and the Birds' Eggs", Lapin graciously shares his eggs with Bouki (Fortier 2011, pp. 30–31). Bouki demands Lapin show him the location of the eggs, and despite Lapin's warnings to not be greedy, Bouki takes too many eggs. As a result, the birds rally against him and nearly rend him to pieces. Finally, "Compair Lapin's Godchild" relays the classic briar-patch story within Fortier's collection (Fortier 2011, pp. 33–35). Lapin manages to eat all of Bouki's butter by following him to his storehouse. When Bouki captures Lapin to punish him, Lapin outwits him and escapes by persuading Bouki to throw him into the briar patch. This being Lapin's home, he escapes. Though critics like Rabalais note that the trickster figure in Louisiana literature is not killed as punishment for violating the social order, the same cannot be said for the 'Bouki the Dupe.' As the dupe, he suffers all manner of pain and humiliation, but he does not die. Conversely, when he attempts to play the trickster or trusts the trickster, Bouki's affairs worsen considerably. The catalog of ill-ends that befall 'Bouki the Trickster'—boiled, beaten, burned, flayed—seem to warn against the dangers of tricksters, in direct contradiction with Rabalais when he writes that "Louisiana Creole tales celebrate the cleverness of the trickster" (Rabalais 2021, p. 29).

'Bouki the Dupe' may not always die, but that does not make the endings of the tales any less morbid. Tellingly, it is never the larger social order that punishes Bouki for his

gullible duplicity when he attempts to play the trickster. Rather, only the individuals who he has harmed punish him. Despite their checkered past, Bouki and Lapin continue to be friends. However, forced together by virtue of the fact they are neighbors in a world without a clear social order, their friendship is strained and contentious.

### 3. Bouki in the Autonomy Chronotope

Unlike the stories told by African-Americans, Bouki goes on to live a second life in the stories of white Creoles, leading Rabalais to conclude that "the social divide across South Louisiana's different communities may have been less marked than in other American states if the shared repertoire of folktales is any indication" (Rabalais 2021, p. 47). According to Wendte, the Autonomous period is aligned with a regional, rural past where fragmented groups of French and Creole speakers developed unique lexical structures and vocabulary. These communities are united in their experience of fracturing; according to Wendte, the shared characteristics are "fractured speech communities... dense communicative networks... and little centralizing energy" (Wendte 2022, p. 103). In this way, local variations of KV were equally confirmed and validated.

The Bouki stories told by white Creoles in the early 20th century clearly exhibit that the dupe of Louisiana folklore meets a more favorable end than the dupe of African legend or myth. The stories under scrutiny here are drawn from Calvin Andre Claudel's *Fools and Rascals: Louisiana Folktales*, a 1979 collection transcribed and translated from French and Kouri-Vini audio files collected during the early part of the 20th century. In these tales, Bouki and Lapin are portrayed less as antagonists and more as friends, in opposition to their friendship in Fortier's earlier stories, where the duo seems thrust into circumstances beyond their control and are now forced to live in close proximity. According to "Bouki and Lapin in the Smokehouse", "They were good friends", such good friends, in fact, that Lapin decides to share with Bouki the secret of how he acquired some "good meat" (Claudel 1979, p. 26). The two visit the Frenchman's smokehouse, where Bouki is undone not by Lapin's treachery but rather by his own gluttony. For this, Bouki is beaten, but he is also given a piece of meat for his troubles—this is Bouki's worst outcome in the collection of tales shared by Claudel. In "The Wine", Lapin manages to drink all of Bouki's wine in an episode reminiscent of the pot of butter in Fortier's collection. Whereas the pot of butter scene ends with the briar-patch scenario, "The Wine" ends with Lapin tricking Bouki again, this time out of the choice bits of the harvest. In "The Princess" Bouki is humiliated when Lapin rides him like a horse. And two versions of the briar-patch story are given, one in "The Tarbaby" and another in "The Magic Door" (Claudel 1979, pp. 30–32). In neither of these versions does Bouki come to an ignoble end.

These last two tales deserve closer scrutiny because they relay a version of the briar-patch story. In "The Tarbaby", Bouki uses a doll made of tar to trap Lapin, sort of like a piece of flypaper, because Lapin has been stealing water from his well. Here, Bouki ties Lapin up rather than using the tar-baby to catch him, but the result is the same: Lapin tricks Bouki into releasing him into the briar patch. In "The Magic Door", the situation is quite different, even if the results are the same. In this tale, Bouki stores his wares behind a magical door sealed with magical words. Bouki catches Lapin in the act of stealing and seals Lapin within the magic door. Lapin uses his wits to be thrown, again, into the briar patch. While the Bouki stories told by white Creoles in the early part of the 20th century would not have been affected by the trauma of slavery, they would have been affected by the legacy of American occupation, which left its own indelible trauma. Since this trauma is historically situated at the contentious locus of language, these stories can be seen as attempts by white Creoles to elaborate on an aspect of their identity they shared with Creoles of color.

By the time these tales reach the white Creoles of Avoyelles Parish, Bouki is just another stock figure divorced from his African roots. In this sense, Bouki can be said to have entered the creolized folkloresque of Louisiana. The folkloresque, to remind the reader, is a corpus of stories and figures from around the world that is available to be consumed as mass

popular culture, regardless of the historically situated origins of those stories and figures. Within the folkloresque, all those stories now find themselves on equal footing—gods and fairies; vampires and angels; ghouls and ghosts. Figures, tropes, types, and situations are mixed and remixed, showing up in contexts in which they might not have originally existed, with multiple characters and situations collapsing and crashing into each other via metaphor and metonymy. Following Jack Zipes, this folkloresque consumption defines the fairy tale, which "continues to grow and embraces, if not swallows, all types of genres, art forms, and cultural institutions; and it adjusts itself to new environments through the human disposition to re-create relevant narratives" (Zipes 2011, p. 222). This description of fairy tales sounds not unlike Mimi Sheller's description of creolization, a concept that is about "moving and mixing elements" but "deeply embedded in situations of coerced transport, racial terror, and subaltern survival" (Sheller 2003, p. 189). If mixing and remixing of this sort happens naturally within the folkloresque, then a creolized folkloresque would exhibit this same sort of eclectic mixing and remixing but within the parameters created by hierarchical power structures such as slavery or linguistic discrimination.

This creolization is more than evident in both the material circumstances and linguistic realities of white Creoles of the early twentieth century. Rabalais emphasizes that the linguistic subtleties of Claudel's stories are lost in the monolingual translations. Listening to the original recording, Rabalais discovered that the original tellers code-switched between French and Kouri-Vini in the original narrations (Rabalais 2021, pp. 44–45). This is unusual because the tales come from Avoyelles Parish, an area where speakers did not adopt as many elements of KV into their everyday French as did some groups in places like St. Martin or St. Landry parishes[7]. At the very end, Lapin's character even code-switches into English. Rabalais interprets this through the lens of narrative and identity, suggesting that white Creoles had taken on a new identity. While that may be true, another way to understand it would be to see that French and KV were subject to similar linguistic pressures during this period in Louisiana. Anyone who spoke English was considered an outsider, but anyone who spoke French or KV would have been considered an insider, even if they might have previously been considered an outsider because of their race. In Louisiana, there have always been those who saw race as secondary to language.[8] In that sense, Lapin's greatest treachery may be to use *English* in a purely francophone/creolophone context.

## 4. Bouki in the Revitalization Chronotope

Bouki stories decreased in popularity throughout the 20th century, no doubt because during this period, most Creoles effectively Americanized. Americanization—identifying as American—becomes prevalent in the wake of WW2 as a result of the patriotic fervor sweeping the nation, no doubt, but in *A Creole Melting Pot: The Politics of Language, Race, and Identity in Southwest Louisiana, 1918–1945*, Christophe Landry shows how the rapid Americanization of Creoles after WW2 was actually the result of a process of integration that has as much to do with Jim Crow laws as WW2. Following Landry, French- and Creole-speaking Louisianans of any color prior to Americanization would have identified as Creole; after Americanization, they see themselves as white Americans or black Americans, and this decision to include the racial modifier is indicative of an American mindset (Landry 2016, pp. 202–43). Along with this Americanization came the loss of French and KV for the majority of speakers in Louisiana, setting the stage for the Revitalization period. Following Wendte, the Revitalization period focused on preserving the language. This is, he notes, paradoxical, because to save the language will require transforming the language (Wendte 2022, pp. 108–9). In this stage, language and identity almost collapsed in on each other. Under this aegis, to be Creole is to speak Creole.

The success of the Revitalization period can be measured through *Contes Merveilleux* by Éditions Tintamarre. These stories mark the first time in history that Louisiana's francophone and creolophone writers produced *literary fairy tales*. This change is monumental for Louisiana—the volume stands as a testimony to the work that has been done over the past decades to revitalize French and KV in Louisiana. But it also represents an opportunity

for Louisiana's writers to think about what their literary fairy tales might look like to the outside world. Unlike the folklore or fairy tales exchanged orally between insular and sometimes even secretive groups, literary fairy tales travel out into the world to encounter an international and educated world populace, becoming part of the larger international folkloresque.

If these literary fairy tales represent the forward guard of Louisiana fairy tales, and if Bouki's story is the story of KV, then language activists should be alarmed by the minor role Bouki plays therein. Among the French-language stories, Bouki is nowhere to be found. In the KV stories, Bouki appears only in a minor role. I refer here to the tale "Dé bon pwason" ("Two Good Fish") by Jonathan Mayers and Henry Johnson, a white Creole and a Creole of color, respectively (Mayers and Johnson 2023)[9]. In this story, Lapin moves to a new village, where he befriends one of the locals, Radbwa, whose name (rat-de-bois) means opossum in North American French. While playing with their pet fish, the pair is set upon by Vyé Séléstin, a three-headed monster. After dispatching the beast, the two find their fish, and they continue on about their happy existence. No one is duped. No one is punished. The tale is entirely sanitized of the more unsavory elements found in previous dupe/trickster tales. In fact, Bouki plays the role of a shop merchant. While the figure of the dupe does not seem like the best to run a shop, perhaps Bouki has learned from his past encounters. Whatever the case, Bouki is warm and welcoming to strangers. According to Lapin, "*Bouki té primiyé moun li trouvé la ki té pa moké sô laksen*/Bouki was the first person he met there who did not mock his accent". Rather than Bouki and Lapin forming an insular group, Bouki must, as a representative of KV, extend an outward and welcoming hand, even to those who may speak a little differently. The threat that KV faces today does not come from slavers or colonizers but from the international dominance of English. As a result of this dominance, *When Languages Die* projects massive language death over the next century (Harrison 2007). KV is in the crosshairs. Perhaps, then, an open, sincere, and inviting stance is the correct one for Bouki to take.

## 5. Bouki at the Crossroads

One solution to the threat of language death might be to figure out ways to market the figures of Louisiana folklore and fairy tales for domestic and international audiences. Rabalais broaches this idea in *Folklore Figures of French and Creole Louisiana*. He endorses the marketing of regional folklore figures such as the *rougarou* "as a means of staking out a regional specificity in opposition to global mass-market brands" (p. 183). While this might be the case, by putting the figures on the "open market", they can and will be appropriated without much concern for the function those figures play in the society from which they came. If anything, regional marketing will only accelerate the rate at which these Louisiana folklore figures enter the larger mosaic of the folkloresque and become commercial products capable of being exploited by those with no historical connections to the community. The *rougarou* might be better known than Bouki because it has been used for regional marketing purposes, yet the term is increasingly associated with a host of stereotypes that restrict the *rougarou* story from fulfilling its true potential as a story of transgression and transformation. Usually, the word *rougarou* is taken to be an adulteration of the French term *loup garou*, which translates as werewolf. The *loup garou* stories were popular in Canada, but the term *rougarou* might signify creolized elements of Native American culture. According to Rabalais, previous scholarship connecting the word *rougarou* to its linguistic roots in *loup garou* "inadvertently serves to obscure the connections between the Louisiana *rougarou* and the Native American oral tradition" (Rabalais 2021, p. 145). This connection is only further obscured by Louisiana French's connection to the international Francophonie, where international speakers are quick to correct *rougarou* as a mispronunciation.[10] Conversely, KV actively resists this impulse, embracing *rougarou* as the proper pronunciation.[11]

KV in Louisiana finds itself at the crossroads. On one hand, KV practitioners can pursue aggressive marketing to sell Kouri-Vini to the widest possible audience, regardless of whether the producers and consumers are actual members of the Creole community.

This would achieve the widest possible dispersion and prove an absolute boon in the fight against language death, but it would mean giving up creative control over figures such as Bouki or the *rougarou*. On the other hand, KV can neglect all outside contact and focus purely on the community, perhaps producing innovative works, but not connecting to the larger folkloresque. KV would atrophy as a result of this lack of contact, even if the language and the stories remained 'authentic.' Perhaps, though, if different communities and cultures created stories together, they could then create new and overlapping identities.

Such new and overlapping identities can be observed in a Youtube video in KV entitled "Lon Zoréy-yé a Lapin", or "Lapin's Long Ears" (Juanga and Guillory-Chatman 2019). In this tale, Bouki and Lapin are friends, neither trying to trick the other; rather, they work together, Bouki freeing Lapin from the grasp of a crab and, in the process, lengthening Lapin's ears, where the crab had pinched him. Written by an African writer, then translated by Adrien Guillory-Chatman, a Creole of color, the ideas in the story emerge from a cultural context far removed from that of modern-day Louisiana but connected through generations by language and story. For Bouki to return home in this way is a testament to his resilience. What's more, the Bouki tales coming from modern-day Louisiana share with the Bouki tales coming from modern-day Africa a similar theme: old antagonists becoming fast friends. Even if the Creolophonie is fragmented, the African diaspora, it appears, is not.

Modern versions of Bouki and Lapin do not depict Bouki and Lapin as contentious friends but rather as true friends. This stance is more appropriate to the language struggles of Louisiana than the more contentious dupe-trickster stories of previous eras. And that makes sense. After all, slavery and colonialism are contentious social structures that require brutal cunning to survive for the enslaved and the colonized. While the potential loss of KV would be devastating culturally for Louisiana, the modern situation in Louisiana could never compare to the unequal power structures of these previous periods. Because linguistic identity is so important in Louisiana, Bouki will continue to have a place in the struggles of Louisiana's peoples, but it seems Bouki will no longer continue to play the role of dupe. The material conditions that created the people who needed the trickster and dupe archetypes have passed, and new material conditions suggest that Bouki and Lapin have work to do as friends to preserve the Louisiana Creole language and culture. Whether those preservation efforts are successful or not, Bouki has, via the literary fairy tale, entered the larger folkloresque, thereby introducing him into the larger world of popular culture. Bouki resonates with members of the African diaspora because he traveled, with them to the New World, via slavery, and back, via modern communication technologies. Bouki resonates with white Creoles because he is a figure of cunning ignorance and duplicitous simplicity. Whether Bouki will resonate with the masses as much as figures such as the *rougarou* remains to be seen, but what remains certain is that, as creators search the folkloresque for inspiration, they will find Bouki available to them, wise and ready to be their friend.

**Funding:** This research received no external funding.

**Data Availability Statement:** No new data were created or analyzed in this study. Data sharing is not applicable to this article.

**Conflicts of Interest:** The author declares no conflict of interest.

## Notes

[1] This couplet is included at the end of Alcée Fortier's "Compair Bouki et Compair Lapin No. 6" in his 1895 collection *Louisiana Folk-tales: Lupin, Bouki, and Other Stories in French Dialect and English Translation*, re-printed in 2011. The Kouri-Vini version here is updated from the original to reflect changing orthography and style in Kouri-Vini. The original translation by Fortier translates *bouki* as goat, which makes sense given the lack of hyenas in Louisiana. This lack has led to "bouki" being translated a number of ways; following Albert Valdman in *The Dictionary of Louisiana Creole* (Valdman 1998), bouki is translated as he-goat, monkey, and fox. However, in *The Dictionary of Louisiana French: As Spoken in Cajun, Creole, and American Indian Communities* (Valdman et al. 2009), bouki is defined solely as a hyena by the editors. Though bouki is, perhaps ahistorically, almost always considered a hyena by modern KV speakers', the word 'critters' was chosen for the translation of this couplet because it is closer to the sense in which Fortier penned the lines *and* for its parallel euphony with 'rabbits' below.

2    The term 'kouri-vini' began as a derogatory term for the language, but is generally accepted as the proper name for Louisiana Creole. Activists prefer this because it avoids the stigma attached to the word 'creole' and establishes KV as a language of it's own, with it's own name for itself. See: https://www.mylhcv.com/languages/2/ (accessed on 22 January 2024).

3    I deliberately write 'the Sale of Louisiana' and not 'the Purchase of Louisiana' to emphasize that while most American commentators might not see the post-1803 period as a 'colonial' period, there is a widely reported sense of having been sold to and occupied by a foreign oppressor (Americans) that degraded and shamed the cultural and lingusitic heritage of the people. In this sense, Louisiana could be said to be a *double* post-colonial state. Perhaps even triple if Spanish rule is considered to be qualitatively different from French rule. Understanding these multiple colonial determinations is essential to understanding the modern Louisiana Creole mentality toward history, culture, and language.

4    Here I cite personal experience. My father did not speak French because my grandparents would spare him and his siblings the humiliation. After French was banned as the language of instruction in Louisiana in 1921, many children were punished corporally for speaking French. My paternal grandfather and grandmother were among this generation. In fact, my grandfather remained illiterate his entire life because of the shame associated with school.

5    *The Economist* concludes "Louisiana Creole is Enjoying a Modest Revival" (*The Economist* 2023), https://www.economist.com/united-states/2023/12/20/louisiana-creole-is-enjoying-a-modest-revival (accessed on 22 January 2024).

6    Ironically, it is during this period that creolization as a metaphor for hybiridity comes into fashion. Lafcadio Hearne, another folklorist with intimate connections to Louisiana, is the first to use creolization in this way. Hearne's use is closer to Glissantian sense outlined above. Hearne, as a white outsider, would not be subject to the unequal power distribution characteristic of the historical use of creolization used in this article. See (Nabae 2014).

7    My own grandfather was born in Avoyelles Parish during this time before eventually moving to New Orleans. I mention this anecdote because this would have been common at the time (circa WW2 he sought employment in the shipyards of New Orleans to escape the penury of picking cotton). In New Orleans, Avoyelles French speakers would have had close contact with New Orleans KV speakers, a fact that no doubt explains the mixture of French and KV spoken in my childhood home. This fact may also indicate that white Creoles in Avoyelles had more experience with KV than scholars have previously thought.

8    This takes two identifiable forms. First, and better documented, Creoles of color often distinguished themselves culturally and linguistically from English-speaking African-Americans; see: (Picone 2003; Barthé 2021). Second, and less documented, many white Creoles treated Creoles of color differently than they treated English-speaking African-Americans. I can cite unsavory elements of family history as paradigmatic of this attitude. One of my grandfather's best friend was a Creole of color. They met in New Orleans, worked together, and often spent time fishing, hunting, or cooking. They delighted in sharing unique turns of phrase or vocabulary. However, my grandfather did not like African-Americans. I have long puzzled over this paradox.

9    This story was also featured in the October 2023 exhibition Mitoloji Latannyèr/Mythologies Louisianaises at the Capitol Park Museum in Baton Rouge, Louisiana.

10   I speak from personal anecdote. I can also attest that, in my own family, the *rougarou* was never portrayed as a werewolf. The rougarou was a scary beast that stalked the swamps, ripping bark from ancient cypresses, but it was not a wolf.

11   The rougarou does appear in my own tale in *Contes Merveilleux* (Cooper and Wendte 2023), but anything more than a footnote seems, to me, gauche. I will let readers decide whether my rendition of the rougarou obscures the connections to the Native American oral tradition of South Louisiana.

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
