# Peer review of "The Secret Lives of Bouki: Louisiana’s Creolized Folkloresque"

_humanities, doi:10.3390/h13010026_

Round 1
Reviewer 1 Report
Comments and Suggestions for Authors
Reviewer's summary
“The Secret Lives of Bouki: The Creolized Fairy Tales of Louisiana” is an interesting, worthwhile article. The ideas and works presented are interested and stand to make an original contribution. I think with a little work the article can be brought up to a very good standard. My main recommendations for revision at the moment:
(1) The article only cites a small range of scholarly literature, drawing heavily on Rabalais (2021). Engaging with more scholarship on Louisiana and African-American folklore will enhance the argumentation of the article. Likewise, parts of the historiography appear underdeveloped or incomplete - to bring out the best in this work, the author needs to engage more closely with prior literature on the Americanization of Creole Louisiana and the revitalization of Kouri-Vini.
(2) In all, parts of the paper are difficult to follow unless the reader is already heavily engaged in the Louisiana context. The flow of the argumentation itself could be much tighter and more logical, possibly even by including section headers but at least by including clear signposting to the reader. If the author does this, it will really help them make their point in an engaging way. An example of this is the author's focus on Bouki stories as told by white Creoles - who is this population? Why are they significant? Why focus on their telling rather than that of Creoles of color? While the answers to these questions may be self-evident to the author, it is difficult as a reader to get a handle on them.
The author could engage with an article which has already appeared in this special issue (DuBois 2023; https://www.mdpi.com/2076-0787/12/5/94) for one example of how to improve their manuscript in line with (1) and (2) above.
Citations at times are unclear or not given; Aarne-Thompson-Uther Index is not given for folktales as is the norm in the field.
I also include here some point-by-point revisions, which I hope will be helpful to the author.
Page-by-page comments
“créolophone” (p. 1) should be "creolophone" (no é in English). Correct this here and throughout.
“settler-slave state” (p. 1) Where is this term from? Should be explained or cited.
“hereafter called Kouri-Vini or KV” (p. 1) Why not give this as "Kouri-Vini" and then "KV" hereafter?
I think this needs to be explained for readers. How does this name differ from 'Louisiana Creole'? Why do you use it instead of 'Louisiana Creole'?
“les ‘Méricains” (p. 1) Italicize this.
“Alcée Fortier’s “Compair Bouki et Compair Lapin No. 6” in his 1895 collection Louisiana Folktales” (p. 1) Page reference? Citation should be matched with the 2011 edition given in the bibliography.
“though Compair Bouki is a hyena” (p. 1) According to whom?
“Picon” (p. 2) Picone
“Bouki” (p. 2) Can you give Bouki a more focused introduction somewhere? It will not be clear to many readers. For an overview, see:
Gaudet, Marcia. 1992. Bouki, the Hyena, in Louisiana and African Tales. The Journal of American Folklore 105(415). 66. DOI: https://doi.org/10.2307/542000
“créolophonie” (p. 2) Italicize if in French, or remove é if in English.
“This author is pleased to be, alongside linguist N.A. Wendte, the winner of the Louisa Lamotte Award for Best Conte Merveilleux in Kouri-Vini.” (p. 2) Can the link between this footnote and the article be clarified?
“Professor Emeritus” (p. 3) Is it necessary to give the title?
“white Creole 102 communities” (p. 3) Why only white communities? These stories have been told also (chiefly?!) by Creoles of Color.
“contes merveilleux” (p. 3) italicize
“In Louisiana stories, Bouki appears alongside Lapin, where he often plays the dupe 106 to Lapin’s trickster.” (p. 3) The introduction of these figures is a little dense. Can an examples be given?
“white Creoles in Louisiana lived under conditions of unequal power reminiscent of the unequal power situations Price argues define creolization as a historical process” (p. 3) I get the point that is being made here, namely that Bouki persists as a folktale character in white communities. However, why is his presence among Creoles of Color not mentioned? It seems to me to make more sense first to develop the argument that Bouki survived in oral literature of the enslaved and their descendants, then mention that he also "goes on to live a second life in the stories of white Creoles and Cajuns". Although it may not be the author's intention, in this section it is difficult to find a place for Bouki in the oral literature of Creoles of Color and that population’s contribution is rendered unclear. The contribution of that population needs to be made clearer. The author needs to help the reader to understand how these stories were transmitted to white communities from communities of color.
For more on Whiteness and orality in this context, the author should consult:
Gipson, Jennifer. 2016. “A Strange, Ventriloquous Voice”: Louisiana Creole, Whiteness, and the Racial Politics of Writing Orality. The Journal of American Folklore 129(514). 459. DOI: https://doi.org/10.5406/jamerfolk.129.514.0459
“no doubt because during this period the number of French and Creole speakers declined precipitously” (p. 3) This argument does not fit well with the following sentence. How can it be that language loss results in the decline of Bouki but the gain in popularity of Rougarou? Couldn't these stories just be told in English anyway (as, I presume, the Rougarou stories are?).
“Rabalais” (p. 3) full citation needed
“Most Creoles and Cajuns effectively Americanize, so resistance in the form 125 of the trickster figures is no longer necessary. Out go Lapin and Bouki, into the briar patch” (p. 3) These sentences read to me as if they are not complete. They can be developed further. "Most Creoles and Cajuns effectively Americanize" holds a lot to unpack for readers unfamiliar with Louisiana.
In my view, this paragraph needs some further elaboration in order to bring the author's argumentation into focus.
“pass the check of some overbearing mother or father, innocuous enough to be re-told by children.” (p. 3) This is a good point !
“near exact” (p. 4) "Near exact" in what sense? Citation needed here for the original version.
“ Whilst he is young,…..” (p. 4) Is this a quote? The formatting should be corrected.
“This shift in worldview is already evident across the stories collected by Fortier, 191 only furthering speculation that the enslaved people might have censored the kinds of 192 stories they told for their young white masters” (p. 4) This section would really benefit from reference to Gipson (2016).
“In most stories featuring Bouki, Bouki plays the dupe to Lapin’s 194 trickster. This is the case in stories “Compair Bouki et Compair Lapin, Nos. 1-6” from part 195 three” (p. 5) This is the kind of explanation that we need when these characters are first introduced.
“Compair Bouki and the Monkey,”” (p. 5) Here and throughout, the page numbers should be given.
“(29)” (p. 5) Check this. Is this a quote? Formatting is off.
“The Tarbaby” (p. 6) Include the Aarne-Thompson-Uther Index, i.e., here: ATU 175, "The Tar-Baby and the Rabbit"
“on par with Jean Sot or Roquelaure” (p. 6) These will not be familiar to readers from outside of Louisiana.
“settled by white Creoles and possessing an educated literate population, as reflected by the use of more standard French constructions found there” (p. 6) This is the first time that the reader can understand why we are focusing specifically on stories as told by white Creoles, who those people are and where their stories derive from. I think this should be integrated much higher up into the introductory paragraphs.
“the two groups were forced together” (p. 6) I think I get what the author is trying to say: Americanization meant that both white Creoles and Creoles of color were othered as non-Americans. But this reads like these two groups became closer as a result. What about racial segregation?! There has been a lot written on this and the author needs to engage with that literature. Surely the transmission of Bouki stories could have happened pre-Americanization, before racial segregation had the Anglo-American binary dynamics; for example in the case of families such as Fortier's.. In general, this article needs to engage more closely with the literature on the history of Creole Louisiana. Some suggestions :
Adams Parham, Angel. 2017. American Routes: Racial Palimpsests and the Transformation of Race. Oxford: Oxford University Press.
Barthé, Darryl. 2021. Becoming American in Creole New Orleans, 1896-1949. Baton Rouge: Louisiana State University Press.
Domínguez, Virginia R. 1986. White By Definition: Social Classification in Creole Louisiana. New Brunswick: Rutgers University Press.
Landry, Christophe. 2016. A Creole melting pot: the politics of language, race, and identity in southwest Louisiana, 1918-45. University of Sussex dissertation. Retrieved from http://sro.sussex.ac.uk/id/eprint/86158/
Trépanier, Cécyle. 1991. The Cajunization of French Louisiana: Forging a Regional Identity. The Geographical Journal 157(2). 161. DOI: https://doi.org/10.2307/635273
Urbain, Emilie. 2017. Hiérarchisation des langues et des locuteurs : différenciation sociale et discours sur la langue dans la francophonie louisianaise depuis la Guerre de Sécession. Revue Transatlantique d’études Suisses 6(7). 199–220.
Waddell, Eric. 1979. La Louisiane française : une poste outre-frontière de l’Amérique française ou un autre pays et une autre culture? Cahiers de géographie du Québec 23(59). 199. DOI: https://doi.org/10.7202/021434ar
“qtd. Price, 2017” (p. 7) Is this Sheller quoted in Price? Why not cite Sheller?
“had taken on a new identity” (p. 7) Had taken on what kind of identity? Are they using KV to index a 'Black voice' (see Gipson 2016)?
“were on equal terms” (p. 7) Or, more precisely, "subject to similar pressures"?
“In Louisiana, there have always 335 been those who saw language as their primary identifier, seeing race as only secondary to 336 that.” (p. 7) Citation most definitely needed!
“Contes Merveilleux” (p. 7) Are these stories referenced here (the author's and others') also counting as stories written by 'white Creoles'? Given the argument so far, identifying the authors of these texts seems significant.
“Unlike the folklore or fairy tales exchanged orally be- 344 tween insular and sometimes even secretive groups, literary fairy tales travel out into the 345 world to encounter an international and educated world populace.” (p. 7) Very nice point.
“lack of Bouki” (p. 7) Wait, this comes from nowhere. So, Bouki is not featured in these stories?
“Dé bon pwason” (p. 8) Interesting discussion of this story.
“entirely sanitized of its more unsavory elements” (p. 8) I don't understand. This implies that there was an original story and that the original version was sanitized in Mayers (2023). This needs to be clarified. Or is the author trying to express that the themes in this modern story are sanitized when compared to earlier iterations?
“Mayers, 2023” (p. 8) Give the citation right away (i.e., when you first mention this story) otherwise it is confusing.
“The threat which Kouri-Vini faces today does 361 not come from slavers or colonizers but from the international dominance of English.” (p. 8) More unpacking needed. Isn't the endangerment of this language just a continuation of the Americanization of Louisiana? The author should engage with some literature on this, at least Dajko (2012).
“Disneyfication” (p. 8) citation for this term?
“Kouri-Vini can neglect all outside contact and focus purely 393 on the community” (p. 9) I think in general the discussion of revitalizing KV could benefit from a more scholarly perspective. If the author wants to engage with these problems, the relevant literature must be cited. For example:
Klingler, Thomas A. 2019. The Louisiana Creole Language Today. In N. Dajko and S. Walton (eds.), Language in Louisiana, 90–107. University Press of Mississippi. DOI: https://doi.org/10.14325/mississippi/9781496823854.003.0007
Mayeux, Oliver. 2022. Language Revitalization, Race, and Resistance in Creole Louisiana. In Rain Prud’homme-Cranford & Darryl Barthé & Andrew Jolivétte (eds.), Louisiana Creole peoplehood: Afro-indigeneity and community, 143–158. Seattle: University of Washington Press. Retrieved from https://www.jstor.org/stable/j.ctv2fjx0c8
Especially helpful here I think would be: Wendte, Nathan A. 2022. The chronotopic organization of Louisiana Creole ethnolinguistic identity. Etudes Francophones 35. 93–121.
““Lon Zoréy-yé a Lapin,” or “Lapin’s Long Ears.”” (p. 9) Citation needed.
“ by an African writer, then translated by Louisiana Creoles” (p. 9) Who? Proper citation needed.
Author Response
Hi,
Thank you for the feedback. I addressed both of your major critiques. I introduced Bouki better, and scaffolded the information in a more accessible way. Regarding research, I added:
Christophe Landry—to explain Americanization
Cćyle Trépanier— to delineate the French subcultures
N.A. Wendte—for the cultural periods of the KV language
Marcie Gaudet—for her previous work on Bouki.
Thomas Klingler—For his summary of the state of things up until 2019.
- Bascom—for the distinction between legend & folklore
Jennifer Gipson—basically what we know from this period is shaped by white voices.
Michael Dylan Foster—the concept of the folkloresque.
I appreciate the effort you put into line by line suggestions, and I’ve done my best to address them all. I hope you will be pleased with the changes.
Reviewer 2 Report
Comments and Suggestions for Authors
Interventions on Lousiana Creolophone are underrepresented and very welcome. Moreover, the author of the piece brings into discuss personal connection and information based on that background.
Generally speaking, substantially increased engagement with contextual scholarship is needed, as the submission relies too heavily on two (high-quality resources) , Rabalais and Zipes. Touchstone research should be engaged (Caryn Cossé Bell, RJ Scott, HE Sterkx, Chris Michaelides) in contextual discussions. This can be done in conjunction with familial/heritage claims (ie fn2: this is widely reported, see e.g. Zachary Richard).
Further discussion ca. 129 of Fortier's original publication, including full original title should be included (scanned archives of the book are available). The fact that AF designates the anthology for professional "folklorists" and "philologists" supports author's claims.
55. Acadian expulsions include other current provinces besides NS.
Citation needed at 119.
Comments on the Quality of English Language
English is generally fine, with the following notes:
61. "some" or "many" appears to missing
101. means
112. appears extra space mark
114. that
124. incomplete citation
143. Conjunction needed between two opposing sentences.
175. Settlers
180-190. fix citation
305. Unclear use of "owners"
Author Response
Hi, Thank you for your feedback. I brought in more contextual discussions as indicated.
Generically, I can give you a broad sense of the research I added:
Christophe Landry—to explain Americanization
Cćyle Trépanier— to delineate the French subcultures
N.A. Wendte—for the cultural periods of the KV language
Marcie Gaudet—for her previous work on Bouki.
Thomas Klingler—For his summary of the state of things up until 2019.
- Bascom—for the distinction between legend & folklore
Jennifer Gipson—basically what we know from this period is shaped by white voices.
Michael Dylan Foster—the concept of the folkloresque.
I accepted your line edits, too.
Reviewer 3 Report
Comments and Suggestions for Authors
All comments and suggestions for the author about content are contained in the attached PDF.

All comments for the author on the quality of the English Language for are contained in the same attached PDF as the one for comments and suggestions about content.
Author Response
Hi,
Thanks for your response. I found it helpful in general. I didn’t care for your reasoning regarding the translation of the opening couplet, but dang it if your translation just wasn’t better poetry. The sounds of critter and rabbit paired together delight me. In general, I’ve accepted your line edits, and I’ve done my best to fix all places where you drew attention to my less than felicitous word choices. I’ve done a lot to improve the organization of the piece, and I have also cleared up the dupe-trickster confusion you highlighted. I do think American colonization ends after WW2 with the completion of the Americanization process. I’m simply offering a view of history from the people who were assed-out by history. Have your own people ever been bought and sold? Anyway, these claims are roughly in-line with similar claims by Bernard and Landry about the Americanization process. I only say the American colonization period is over now because most Creoles are Americanized (again, see Landry). As per the essay, I’ve done my best to clarify my language and be more precise. We’ll see what you think I suppose..
Generically, I can give you a broad sense of the research I added:
Christophe Landry—to explain Americanization
Cćyle Trépanier— to delineate the French subcultures
N.A. Wendte—for the cultural periods of the KV language
Marcie Gaudet—for her previous work on Bouki.
Thomas Klingler—For his summary of the state of things up until 2019.
Bascom—for the distinction between legend & folklore
Jennifer Gipson—basically what we know from this period is shaped by white voices.
Michael Dylan Foster—the concept of the folkloresque.
Reviewer 4 Report
Comments and Suggestions for Authors
This is a very good paper that I feel deserves publication, but there are parts of the argument that might need further work. One key problem of the paper is that the author, following Rabalais perhaps, seems to suggest that the folktales under consideration were rendered in some way suggestive of abject states of superstition following the conversion of the Africans to Christianity, leading to a difference between animal tales of Africa and those of Louisiana:
Rabalais goes on to recognize the fundamentally different worldview found in Louisiana stories, that in Louisiana, “animal tales seem to exist in a spatial and temporal frame removed and independent from the real world,” unlike the African tales, where the figures are integrated into reality itself (Rabalais, 2021, 162 p. 33).
This division is the opposition (see Bascom’s early classification here, Bascom, W., 1965. The forms of folklore: Prose narratives. The Journal of American Folklore, 78(307), pp.3-20.)between legend (tales told as true) and folktale (tales told as fictions, not related to our shared reality). It is not clear to me then why folktales, which point to no “belief” whatsoever, could attract the charge of “superstition” after conversion, as stated here:
Once the enslaved African had assimilated into Christianity, their original folktales 186 would have been immediately quarantined as “superstitions” that could be dangerous to 187 children. Even though trickster stories continued to be told, their reduced social function 188 resulted from the trauma of enslavement, yes, but also from the demotion of the super- 189 natural everywhere in the Western world.
Folktales are not legends, and only legends or myths can attract charges of superstition, since folktales do not entail belief, quite the contrary! In fact, Bascom notes that converting an erstwhile “true” story (myth or legend) into a folktale (a tale that is decidedly told as being a complete fiction, like Uncle Remus or the comedian Dolemite’s reworkings of African diasporic folktales) is a common result of the underlying ontology of cosmology falling apart, leaving an erstwhile myth or legend told now as a children’s story. As Bascom notes, fairy tales are without exception legends, and only take on the properties listed by the author after they are demoted from true, folkloric stories to untrue, literary children’s stories in the 19th century, which adequately explains the differences noted by the author between folkloric folktales and literary fairy tales (Rabalais says this, but it is not that fairy tales are relatively literary, they are literary in origin, even if, like folkloresque creepypasta monsters like Slenderman, or gothic haunted house narratives from gothic novels, they end up in the “folkloresque moebius strip” and start circulating as actual folklore: the gothic novel spawns the gothic haunted house narrative of legend, for example. I think Zipes porous fairy tale really needs to be reconceptualized within the framework of the folkloresque, but in themselves, in whatever medium, fairy tales are folktales and not legends: I can tell a story about a house haunted by ghosts and set it nearly anywhere as a legend, a tale told as true, but if I replace ghosts with fairies, hilarity ensues, because fairies have been demoted from real to unreal, from legend to folktale. I don’t think telling children’s stories is necessarily plausibly readable as “superstition. The author appears to attend, in part, to this transition from folkloric to folkloresque. Examples of this porous boundary between folklore and folkloresque abound:
Foster, M.D. and Tolbert, J.A., 2015. The Folkloresque. University Press of Colorado.
Blank, T.J. and McNeill, L.S., 2018. Slender man is coming: Creepypasta and contemporary legends on the internet. University Press of Colorado.
It is indeed the case that for example, fairy stories of the British isles, which begin as legends (and some people claim are fragments of an erstwhile mythology) in the 19th century either turn into more “plausible” ghost stories (and indeed almost all fairies in the new world become ghosts as well see James) or they become largely literary folktales (see Manning) This is a familiar process where the original legend (indexing possible superstition) becomes either a more ontologically acceptable kind of legend (a fairy becomes a ghost) or becomes a story with no ontological commitments (fairy legends become fairy folktales printed in Childrens’ book). Demotion of the supernatural should only affect legends and myths, not folktales, the telling of which does not, prima facie, imply any beliefs at all of the supernatural variety. It is possible that they would be regarded as suspect nonsense, an idle waste of time perhaps that indicates the person is not in some other way not a respectable Christian, after conversion, but that is not at all the same thing as being regarded as superstition.
For example in the following e see some of these transitions
James, R.M., 1992. Knockers, knackers, and ghosts: immigrant folklore in the Western mines. Western Folklore, 51(2), pp.153-177.
Manning, P., 2016. Pixies' Progress. The Folkloresque: Reframing Folklore in a Popular Culture World, p.81.
If the explanation isn’t based on a sense that telling animal stories indexes a superstitious “backwards” non-Christian mind (which seems like an argument that needs a lot more force to be successful), perhaps the abjection of these folktales results, as the author suggests elsewhere, from the original contexts with which they are associated, namely, this suggestion, an argument which is completely different from the superstition argument, namely that:
More telling, Fortier admits that Creoles of color were not eager to share stories with him: “It is a strange fact that the old negroes do not like to relate those tales with which they enchanted the little masters before the war” (Fortier, 2011, p. xvii). If the trickster stories such as the tar baby were folkloric retellings shaped by the pressures of enslavement, then it should not be surprising to suggest the tellers related to the tales differently once slavery ended.
This is actually, to my mind, although it comes from a problematic source, a more successful argument, and it is different, which is that it has nothing to do with folktales indicating “backwardness” of superstition, but rather the tales are strongly associated with a colonial context of slavery, or the way they were told to their masters also in some way ruined these narratives for retellings, since they index “the trauma of slavery”. For tellers of folkloric legends in Greece, as Stewart (demons and the devil) argues, it is not so much the fact that these tales index superstitions that is the problem, it is that they index a “peasant” or “rural” identity, which is, for lack of a better word, “cringe”:
Stewart, C., 1989. Hegemony or rationality?: the position of the supernatural in modern Greece. Journal of modern Greek studies, 7(1), pp.77-104.
I add in passing that I believe the first person to use creolisation in its contemporary sense was folklorist Lafcadio Hearn, who worked in Louisiana:
Bronner, S.J., 2005. " Gombo" folkloristics: Lafcadio Hearn's creolization and hybridization in the formative period of folklore studies. Journal of Folklore Research, 42(2), pp.141-184.
Nabae, H., 2014. Creolization in Lafcadio Hearn’s New Orleans and Martinique Writings. Review of International American Studies, 7(1), pp.131-150.
Author Response
Hi,
Thank you very much for this feedback. I did not know about the folkloresque, but after learning about it, I feel as if my argument would be impossible without the concept. I wish I had known about the concept when I wrote my dissertation in 2011, because it would have helped me make some links between fantasy and folklore that I struggled to make there. That, and I was not aware of Bascom’s distinctions. Also very helpful. I think once you read the article you will be pleased with the changes.
Generically, I can give you a broad sense of the research I added:
Christophe Landry—to explain Americanization
Cćyle Trépanier— to delineate the French subcultures
N.A. Wendte—for the cultural periods of the KV language
Marcie Gaudet—for her previous work on Bouki.
Thomas Klingler—For his summary of the state of things up until 2019.
Bascom—for the distinction between legend & folklore
Jennifer Gipson—basically what we know from this period is shaped by white voices.
Michael Dylan Foster—the concept of the folkloresque.
In general, I accepted most line edits that were offered, too.
Round 2
Reviewer 2 Report
Comments and Suggestions for Authors
Thorough additions now provide a scholarly historical overview, giving key context to the presentation.
Comments on the Quality of English LanguageLanguage is largely quite clear.
Author Response
Thank you for your labor.
Reviewer 3 Report
Comments and Suggestions for Authors
The organization and flow of this article is much improved. See comments about minor revisions to attend to. The comments are attached as notes to highlighted text in the uploaded file.
This article is now very close to being ready to publish, once the minor revisions are taken care of.

The wording has also improved. A scattering of infelicities remain. See the uploaded document.
Author Response
I've made the changes you suggested. I think you will find that one footnote a little more rigorous. I enjoyed this work, it's been fun. Thank you for your labor.